

# High throughput resource efficient reconfigurable interleaver for MIMO WLAN application

Bijoy Kumar Upadhyaya[1], Pijush Kanti Dutta Pramanik[2] and Salil Kumar Sanyal[3]

[1] Department of Electronics & Communication Engineering, Tripura Institute of Technology, Narsingarh, Tripura, India
[2] Department of Computer Science & Engineering, National Institute of Technology, Durgapur, West Bengal, India
[3] Department of Electronics & Telecommunication Engineering, Jadavpur University, Kolkata, West Bengal, India

Corresponding author
Pijush Kanti Dutta Pramanik,
pijushjld@yahoo.co.in

## ABSTRACT

Demand for high-speed wireless broadband internet service is ever increasing. Multiple-input-multiple-output (MIMO) Wireless LAN (WLAN) is becoming a promising solution for such high-speed internet service requirements. This paper proposes a novel algorithm to efficiently model the address generation circuitry of the MIMO WLAN interleaver. The interleaver used in the MIMO WLAN transceiver has three permutation steps involving floor function whose hardware implementation is the most challenging task due to the absence of corresponding digital hardware. In this work, we propose an algorithm with a mathematical background for the address generator, eliminating the need for floor function. The algorithm is converted into digital hardware for implementation on the reconfigurable FPGA platform. Hardware structure for the complete interleaver, including the read address generator and memory module, is designed and modeled in VHDL using Xilinx Integrated Software Environment (ISE) utilizing embedded memory and DSP blocks of Spartan 6 FPGA. The functionality of the proposed algorithm is verified through exhaustive software simulation using ModelSim software. Hardware testing is carried out on Zynq 7000 FPGA using Virtual Input Output (VIO) and Integrated Logic Analyzer (ILA) core. Comparisons with few recent similar works, including the conventional Look-Up Table (LUT) based technique, show the superiority of our proposed design in terms of maximum improvement in operating frequency by 196.83%, maximum reduction in power consumption by 74.27%, and reduction of memory occupancy by 88.9%. In the case of throughput, our design can deliver 8.35 times higher compared to IEEE 802.11n requirement.

# INTRODUCTION

The increasing use of multimedia services and the growth of graphics-based web contents have escalated the demand for high-speed wireless broadband communications. The use of

more than one antenna at the transmitter and/or at the receiver aims to substantially improve the transmission/reception rate. Orthogonal Frequency Division Multiplexing (OFDM) is becoming a popular technique for high data rate wireless transmission (*Yang, 2005*). OFDM may be combined with multiple antennas at both the access point and the mobile terminal to increase diversity gain and/or to enhance the system capacity on a time-varying multipath fading channel, resulting in a Multiple-Input-Multiple-Output (MIMO) OFDM system (*Bolcskei, 2006*; *Hwang et al., 2009*).

The IEEE 802.11n, an amendment to the IEEE 802.11 standard, is based on MIMO-OFDM transmission techniques to enable high-speed data communication with a maximum throughput of 600 Mbps (*Paul & Ogunfunmi, 2008*). In high-throughput wireless communication systems, interleavers (*Niu, Ouyang & Ngo, 2006*) play an important role in reducing the effect of a burst error in the channel and improve the performance of Forward Error Correction (FEC) techniques.

In general terms, an interleaver consists of two parts: address generator and interleaver memory. The address generation process in MIMO WLAN transceiver is implementing the three steps in permutation wherein a floor function is involved (*IEEE 802.11n-2009, 2009*). But due to the non-availability of digital hardware in practice, it is difficult to implement the floor function. Due to this issue, the Look-up Table (LUT) based approach is generally used (*Asghar & Liu, 2009*). In a LUT-based approach, all possible addresses are pre-calculated and stored in the memory. The LUT-based technique is, in general, unattractive (*Upadhyaya & Sanyal, 2013*) as it requires a large number of memory blocks (LUTs) to house the addresses associated with different permissible modulation schemes, bandwidths (BWs), and spatial streams. In addition, the LUT-based address generator requires a large memory access time resulting in slower operation.

Literature review for MIMO WLAN transceiver implementation on hardware platform unveils some works. A work reported in *Eickhoff et al. (2011)* demonstrated the development of a prototype transceiver for IEEE 802.11a, followed by the upgradation to $1 \times 4$ MIMO WLAN. The authors claimed to implement the transceiver on Xilinx FPGA Virtex V LX330. *Setiawan et al. (2011)* demonstrated prototyping of the $2 \times 2$ MIMO WLAN system using Register Transfer Level (RTL) design. The authors used the Model-Based Design Process (MBDP) for developing the RTL design of the transceiver and implemented on Altera FPGA Stratix-II EP2S180 and obtained maximum throughput of 144 Mbps. Another work for MIMO-OFDM transceiver implementation on Xilinx Virtex IV FPGA has been reported in *Srinandhini & Vaithianathan (2014)*. Here, the authors implemented an interleaver and de-interleaver pair, including the convolutional channel coding algorithm of the MIMO-OFDM transceiver. This paper focuses on implementing the fundamental interleaving technique, which does not include inter-row, inter-column permutation, and frequency rotation parameters essential for the MIMO-OFDM transceiver. The FPGA implementation works presented in *Eickhoff et al. (2011)*, *Setiawan et al. (2011)* and *Srinandhini & Vaithianathan (2014)* were neither aimed at full $4 \times 4$ MIMO-OFDM implementation nor achieving the 600 Mbps throughput target. ASIC implementations of the MIMO-OFDM/IEEE 802.11n transceiver are described by some researchers in *Perels et al. (2005)* and *Tran et al. (2012)*. Another recent

work on designing an FPGA-based address generator for a multi-standard interleaver is reported in the literature (*Babu & Gopalakrishnan, 2019*). The authors made a combined implementation of the address generator for WLAN (802.11a/b/g), WiMAX, and 3GPP LTE, but not of the MIMO WLAN (802.11n).

However, the implementations of *Eickhoff et al. (2011)*, *Setiawan et al. (2011)*, *Srinandhini & Vaithianathan (2014)*, *Perels et al. (2005)*, *Tran et al. (2012)* and *Babu & Gopalakrishnan (2019)* are not specifically focused on interleaver/de-interleaver. They do not contain detailed implementation results leaving scope for design optimization with respect to resource utilization, providing compact design, resulting in higher throughput and reduced power consumption.

Very few papers reporting the hardware implementation of the MIMO WLAN interleaver are available in the literature. *Zhang et al. (2009)* presented a de-interleaver address generator implementation on a 0.13 μm CMOS platform. The authors claimed that the implementation was also done on the FPGA platform but without any implementation result. 2-D translation of the interleaver equations for hardware simplicity was proposed in *Asghar & Liu (2009)*. The final expressions, so derived, are very complex and do not clearly explain the hardware design issues, especially for 64-QAM. The implementation platform of this work is reported to be 65 nm CMOS technology. Another recent work (*Zhang et al., 2012*) reported by the authors of *Zhang et al. (2009)* claimed betterment over their previous work in reducing complexity and improvement in maximum operating frequency keeping the same implementation platform. The improvement claimed by the authors is due to exchanging steps between the interleaver and the de-interleaver. In *Kim & Kim (2017)*, the authors presented an FPGA-based implementation of the complete MIMO PHY modulator for IEEE 802.11n WLAN. The implementation is further extended to ASIC with 65 nm CMOS technology. The authors tabulated the FPGA and ASIC implementation results of the complete MIMO PHY modulator for IEEE 802.11n WLAN without mentioning the interleaver's resource occupancy or power consumption separately.

The above-mentioned issues have opened up further research scope for improving the implementation of the MIMO WLAN interleaver that complies with the high throughput data transmission requirements. In this paper, we propose a novel design of the interleaver used in a $4 \times 4$ MIMO WLAN transceiver. The proposed address generator algorithm eliminates the requirement of floor function from the address generator of the MIMO WLAN interleaver.

The key contributions of this work are:

- The mathematical modeling of the new algorithm has been derived with general validity.
- The novel address generator algorithm has been generalized to accommodate more modulation schemes if required.
- The complete MIMO WLAN interleaver, including the proposed address generator algorithm, is transformed into digital hardware and implemented on Spartan 6 FPGA (*Xilinx, 2015*) using Xilinx ISE 12.1.

- To reduce the resource and power consumption and to enhance the throughput, the embedded resources of FPGA like dual-port Block RAM (*Xilinx, 2011*) and DSP blocks (DSP48A1) (*Xilinx, 2009*) have been successfully interfaced and utilized in the hardware model. This approach makes the design very compact and highly efficient. Comparison with existing similar works endorses the superiority of our proposed design in terms of multiple FPGA parameters.
- The functionality test of the address generator has been verified using ModelSim XE-III software.
- Further, hardware testing of the algorithm has also been carried out using Virtual Input and Output (VIO) and Integrated Logic Analyser (ILA) module on Zynq 7000 FPGA board, which further validates the proposed algorithm.
- The proposed design has been compared with a few recent implementations (*Asghar & Liu, 2009*; *Zhang et al., 2009*; *Zhang et al., 2012*) by converting them into FPGA equivalent implementation using *Kuon & Rose (2006)*. The comparison shows the superiority of the proposed design in terms of operating frequency, power consumption, and memory occupancy.
- The performance of the proposed interleaver is compared with IEEE 802.11n. The comparison results show that the proposed interleaver delivers much higher throughput than the maximum throughput requirement of IEEE 802.11n.

The rest of the paper is organized as follows. "Interleaving in IEEE 802.11n" presents the theoretical background of the interleaving process in the MIMO WLAN transceiver. "Proposed Algorithm for Address Generator of Interleaver" presents the proposed algorithm, including the mathematical background for the address generator. A description of the transformation of the proposed algorithm into digital hardware has been made in "Transformation into Hardware". Simulation results followed by FPGA implementation details have been reported in "Simulation Results" and "FPGA Implementation Results", respectively. The concluding remarks are given in "Conclusions".

## INTERLEAVING IN IEEE 802.11N

In a MIMO WLAN transceiver, the encoded data stream obtained from the convolutional encoder is fed to a special type of block interleaver. Interleaving in 802.11n is a three-step process in which the first two steps provide spatial interleaving, and the final step performs frequency interleaving (*Paul & Ogunfunmi, 2008*). The interleaving steps are defined in the form of three blocks shown in Fig. 1. The first step ($B_1$) ensures that adjacent coded bits are mapped onto non-adjacent subcarriers, while the second step ($B_2$) is responsible for the mapping of adjacent coded bits alternately onto less or more significant bits of the constellation, thus avoiding long runs of lowly reliable bits. If more than one spatial stream exists in the 802.11n physical layer, the third step, called frequency rotation ($B_3$), will be applied to the additional spatial streams. The frequency rotation ensures that the consecutive carriers used across spatial streams are not highly correlated.

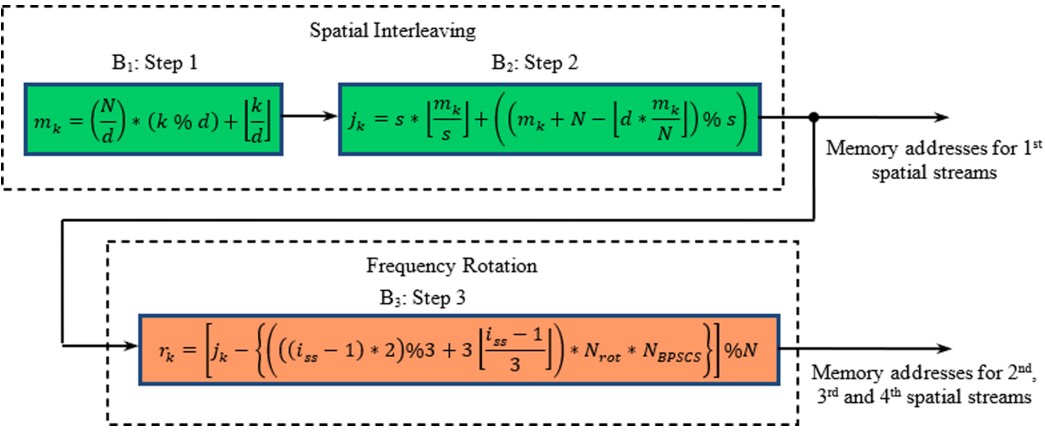

**Figure 1 Block diagram of steps involved in interleaving process for MIMO WLAN.**

Here, N is the block size corresponding to the number of coded bits per allocated sub-channels per OFDM symbol. d represents the number of columns in the interleaver, whose values are 13 and 18 for 20 and 40 MHz BW (*Niu, Ouyang & Ngo, 2006*), respectively. The parameter s is defined as s = max (1, $N_{BPSCS}$), whereas $N_{BPSCS}$ is the number of coded bits per sub-carrier and takes the values 1, 2, 4, and 6 for BPSK, QPSK, 16-QAM, and 64-QAM, respectively. $i_{ss}$ is the index of the spatial stream, and $N_{rot}$ is the parameter used for defining different rotations for the 20 and 40 MHz cases. The operator % and $\lfloor \ \rfloor$ represent modulo function and floor function, respectively.

## PROPOSED ALGORITHM FOR ADDRESS GENERATOR OF INTERLEAVER

The permutation steps described in the $B_1$, $B_2$, and $B_3$ blocks of Fig. 1 involve floor function. Our objective is to propose a hardware-friendly algorithm to implement the address generator on a reconfigurable platform that involves no floor function. Initially, a MATLAB program is developed by implementing $B_1$, $B_2$, and $B_3$ blocks of Fig. 1 to determine the interleaver addresses for all modulation schemes, spatial streams, and BWs. Tables 1(A)–(C) show selected part these addresses for $N_{bpscs} = 1$, $N = 52$, $i_{ss} = 4$; $N_{bpscs} = 4$, $N = 208$, $i_{ss} = 2$; and $N_{bpscs} = 6$, $N = 312$, $i_{ss} = 3$ with 20 MHz BW. Careful examination of these addresses reveals the correlation among them, which may be proposed to express by new algorithms, as described in Tables 2(A)–(C). The complete mathematical formulations of the proposed algorithms, including all modulation schemes, spatial streams, and BWs, are represented by Eqs. (1)–(3).

$$k_{n(QPSK-BPSK)}$$
$$= \begin{cases} D*(i+I)+(j+J) & \text{when } j<(D-J) \text{ and } i<(C-I) \\ D*\{i-(C-I)\}+(j+J) & \text{when } j<(D-J) \text{ and } i \geq (C-I) \\ D*(i+I+1)+\{j-(D-J)\} & \text{when } j \geq (D-J) \text{ and } i<(C-I-1) \\ D*\{i-(C-I-1)\}+\{j-(D-J)\} & \text{when } j \geq (D-J) \text{ and } i \geq (C-I-1) \end{cases} \quad (1)$$

**Table 1A** Part of interleaver write addresses with $N_{bpscs} = 1$, $N = 52$, $i_{ss} = 4$, $BW = 20$ MHz

| Row no (j) | Column no (i) | | | | | | | |
|---|---|---|---|---|---|---|---|---|
| | **0** | **1** | **2** | **...** | **9** | **10** | **11** | **12** |
| 0 | 13 | 17 | 21 | ... | 49 | 1 | 5 | 9 |
| 1 | 14 | 18 | 22 | ... | 50 | 2 | 6 | 10 |
| 2 | 15 | 19 | 27 | ... | 51 | 3 | 7 | 11 |
| 3 | 16 | 20 | 28 | ... | 0 | 4 | 8 | 12 |

**Table 1B** Part of interleaver write addresses with $N_{bpscs} = 4$, $N = 208$, $i_{ss} = 2$, $BW = 20$ MHz

| Row no (j) | Column no (i) | | | | | | | | |
|---|---|---|---|---|---|---|---|---|---|
| | **0** | **1** | **2** | **...** | **6** | **7** | **8** | **...** | **12** |
| 0 | 104 | 121 | 136 | ... | 200 | 9 | 24 | ... | 88 |
| 1 | 105 | 120 | 137 | ... | 201 | 8 | 25 | ... | 89 |
| 2 | 106 | 123 | 138 | ... | 202 | 11 | 26 | ... | 90 |
| ... | ... | ... | ... | ... | ... | ... | | ... | ... |
| 7 | 111 | 126 | 143 | ... | 207 | 14 | 31 | ... | 95 |
| 8 | 112 | 129 | 144 | ... | 0 | 17 | 32 | ... | 96 |
| 9 | 113 | 128 | 145 | ... | 1 | 16 | 33 | ... | 97 |
| 10 | 114 | 131 | 146 | ... | 2 | 19 | 34 | ... | 98 |
| ... | ... | ... | ... | ... | ... | ... | | ... | ... |
| 15 | 119 | 134 | 151 | ... | 7 | 22 | 39 | ... | 103 |

**Table 1C** Part of interleaver write addresses with $N_{bpscs} = 6$, $N = 312$, $i_{ss} = 3$, $BW = 20$ MHz

| Row no (j) | Column no (i) | | | | | | | | |
|---|---|---|---|---|---|---|---|---|---|
| | **0** | **1** | **2** | **3** | **4** | **5** | **6** | **...** | **12** |
| 0 | 234 | 260 | 283 | 306 | 20 | 43 | 66 | ... | 210 |
| 1 | 235 | 258 | 284 | 307 | 18 | 44 | 67 | ... | 211 |
| 2 | 236 | 259 | 282 | 308 | 19 | 42 | 68 | ... | 212 |
| ... | ... | ... | ... | ... | ... | ... | | ... | ... |
| 5 | 239 | 262 | 285 | 311 | 22 | 45 | 71 | ... | 215 |
| 6 | 240 | 266 | 289 | 0 | 26 | 49 | 72 | ... | 216 |
| 7 | 241 | 264 | 290 | 1 | 24 | 50 | 73 | ... | 217 |
| 8 | 242 | 265 | 288 | 2 | 25 | 48 | 74 | ... | 218 |
| ... | ... | ... | ... | ... | ... | ... | ... | ... | ... |
| 23 | 257 | 280 | 303 | 17 | 40 | 63 | 89 | ... | 233 |
**Table 2A  Proposed algorithm for $N_{bpscs}$ = 1 or 2 (BPSK/QPSK) with all $N$, $i_{ss}$ and $BW$**

| Row no. ($j$) | | Column no. ($i$) | | |
|---|---|---|---|---|
| | | 0, 1, 2, 3, … | … | …, C-4, C-3, C-2, C-1 |
| 0, 1, 2, … | $j < (D − J)$ | $i < (C − I)$ <br> $D * (i + I) + (j + J)$ | … | $i >= (C − I)$ <br> $D * \{i − (C − I)\} + (j + J)$ |
| …D-3, D-2, D-1 | $j >= (D − J)$ | $i < (C − I − 1)$ <br> $D * (i + I + 1) + \{j − (D − J)\}$ | … | $i >= (C − I − 1)$ <br> $D * \{i − (C − I − 1)\} + \{j − (D − J)\}$ |

**Table 2B  Proposed algorithm for $N_{bpscs}$ = 4 (16-QAM) with all $N$, $i_{ss}$ and $BW$**

| Row no. ($j$) | | Column no. ($i$) | | | | |
|---|---|---|---|---|---|---|
| | | 0, 1, 2, 3, … | | … | …, C-4, C-3, C-2, C-1 | |
| | | $\{i < (C − I)\}$ & $(i\% 2 = 0)$ | $\{i < (C − I)\}$ & $(i\% 2 = 1)$ | … | $\{i >= (C − I)\}$ & $(i\% 2 = 0)$ | $\{i >= (C − I)\}$ & $(i\% 2 = 1)$ |
| 0, 1, 2, … | $\{j < (D − J)\}$ & $(j\% 2 = 0)$ | $D * (i + I) + (j + J)$ | $D * (i + I) + (j + J + 1)$ | … | $D * \{i − (C − I)\} + (j + J)$ | $D * (i − (C − I)) + (j + J + 1)$ |
| | $\{j < (D − J)\}$ & $(j\% 2 = 1)$ | $D * (i + I) + (j + J)$ | $D * (i + I) + (j + J − 1)$ | | $D * (i − (C − I)) + (j + J)$ | $D * (i − (C − I)) + (j + J − 1)$ |
| | | $\{i < (C − I − 1)\}$ & $(i\% 2 = 0)$ | $\{i < (C − I − 1)\}$ & $(i\% 2 = 1)$ | | $\{i >= (C − I − 1)\}$ & $(i\% 2 = 0)$ | $\{i >= (C − I − 1)\}$ & $(i\% 2 = 1)$ |
| …, D-3, D-2, D-1 | $\{j >= (D − J)\}$ & $(j\% 2 = 0)$ | $D * (i + I + 1) + \{j − (D − J)\}$ | $D * (i + I + 1) + \{j − (D − J − 1)\}$ | … | $D * \{i − (C − I − 1)\} + \{j − (D − J)\}$ | $D * \{i − (C − I − 1)\} + \{j − (D − J − 1)\}$ |
| | $\{j >= (D − J)\}$ & $(j\% 2 = 1)$ | $D * (i + I + 1) + (j − (D − J))$ | $D * (i + I + 1) + \{j − (D − J + 1)\}$ | | $D * \{i − (C − I − 1)\} + \{j − (D − J)\}$ | $D * \{i − (C − I − 1)\} + \{j − (D − J + 1)\}$ |

**Table 2C  Proposed algorithm for $N_{bpscs}$ = 6 (64−QAM) with all $N$, $i_{ss}$ and $BW$**

| Row no. ($j$) | | Column no. ($i$) | | | | | | |
|---|---|---|---|---|---|---|---|---|
| | | 0, 1, 2, 3, … | | | … | …, C-4, C-3, C-2, C-1 | | |
| | | $\{i < (C − I)\}$ & $(i\% 3 = 0)$ | $\{i < (C − I)\}$ & $(i\% 3 = 1)$ | $\{i < (C − I)\}$ & $(i\% 3 = 2)$ | $\{i >= (C − I)\}$ & $(i\% 3 = 0)$ | $\{i >= (C − I)\}$ & $(i\% 3 = 1)$ | $\{i >= (C − I)\}$ & $(i\% 3 = 2)$ |
| 0, 1, 2, … | $\{j < (D − J)\}$ & $(j\% 3 = 0)$ | $D * (i + I) + (j + J)$ | $D * (i + I) + (j + J + 2)$ | $D * (i + I) + (j + J + 1)$ | $D * \{i − (C − I)\} + (j + J)$ | $D * \{i − (C − I)\} + (j + J + 2)$ | $D * \{i − (C − I)\} + (j + J + 1)$ |
| | $\{j < (D − J)\}$ & $(j\% 3 = 1)$ | $D * (i + I) + (j + J)$ | $D * (i + I) + (j + J − 1)$ | $D * (i + I) + (j + J + 1)$ | $D * \{i − (C − I)\} + (j + J)$ | $D * \{i − (C − I)\} + (j + J − 1)$ | $D * \{i − (C − I)\} + (j + J + 1)$ |
| | $\{j < (D − J)\}$ & $(j\% 3 = 2)$ | $D * (i + I) + (j + J)$ | $D * (i + I) + (j + J − 1)$ | $D * (i + I) + (j + J − 2)$ | $D * \{i − (C − I)\} + (j + J)$ | $D * \{i − (C − I)\} + (j + J − 1)$ | $D * \{i − (C − I)\} + (j + J − 2)$ |
| | | $\{i < (C − I − 1)\}$ & $(i\% 3 = 0)$ | $\{i < (C − I − 1)\}$ & $(i\% 3 = 1)$ | $\{i < (C − I − 1)\}$ & $(i\% 3 = 2)$ | $\{i >= (C − I − 1)\}$ & $(i\% 3 = 0)$ | $\{i >= (C − I − 1)\}$ & $(i\% 3 = 1)$ | $\{i >= (C − I − 1)\}$ & $(i\% 3 = 2)$ |
| …, D-3, D-2, D-1 | $\{j >= (D − J)\}$ & $(j\% 3 = 0)$ | $D * (i + I + 1) + \{j − (D − J)\}$ | $D * (i + I + 1) + \{j − (D − J − 2)\}$ | $D * (i + I + 1) + \{j − (D − J − 1)\}$ | $D * \{i − (C − I − 1)\} + \{j − (D − J)\}$ | $D * (i − (C − I − 1)) + (j − (D − J − 2))$ | $D * (i − (C − I − 1)) + (j − (D − J − 1))$ |
| | $\{j >= (D − J)\}$ & $(j\% 3 = 1)$ | $D * (i + I + 1) + (j − (D − J))$ | $D * (i + I + 1) + (j − (D − J + 1))$ | $D * (i + I + 1) + (j − (D − J − 1))$ | $D * (i − (C − I − 1)) + (j − (D − J))$ | $D * (i − (C − I − 1)) + (j − (D − J + 1))$ | $D * (i − (C − I −1)) + (j − (D − J − 1))$ |
| | $\{j >= (D − J)\}$ & $(j\% 3 = 2)$ | $D * (i + I + 1) + (j − (D − J))$ | $D * (i + I + 1) + (j − (D − J + 1))$ | $D * (i + I + 1) + (j − (D − J + 2))$ | $D * (i − (C − I − 1)) + (j − (D − J))$ | $D * (i − (C − I − 1)) + (j − (D − J + 1))$ | $D * (i − (C − I − 1)) + (j − (D − J + 2))$ |

$k_{n(16-QAM)}$

$$= \begin{cases}
D*(i+I)+(j+J) & \text{when } \{j<(D-J)\}\&[\{i<(C-I)\}\&(i\%2=0)] \\
D*(i+I)+(j+J+1) & \text{when } [\{j<(D-J)\}\&(j\%2=0)]\&[\{i<(C-I)\}\&(i\%2=1)] \\
D*\{i-(C-I)\}+(j+J) & \text{when } \{j<(D-J)\}\&[\{i\geq(C-I)\}\&(i\%2=0)] \\
D*(i-(C-I))+(j+J+1) & \text{when } [\{j<(D-J)\}\&(j\%2=0)]\&[\{i\geq(C-I)\}\&(i\%2=1)] \\
D*(i+I)+(j+J-1) & \text{when } [\{j<(D-J)\}\&(j\%2=1)]\&[\{i<(C-I)\}\&(i\%2=1)] \\
D*(i-(C-I))+(j+J-1) & \text{when } [\{j<(D-J)\}\&(j\%2=1)]\&[\{i\geq(C-I)\}\&(i\%2=1)] \\
D*(i+I+1)+\{j-(D-J)\} & \text{when } \{j\geq(D-J)\}\&[\{i<(C-I-1)\}\&(i\%2=0)] \\
D*(i+I+1)+\{j-(D-J-1)\} & \text{when } [\{j\geq(D-J)\}\&(j\%2=0)]\&[\{i<(C-I-1)\}\&(i\%2=1)] \\
D*\{i-(C-I-1)\}+\{j-(D-J)\} & \text{when } \{j>=(D-J)\}\&[\{i\geq(C-I-1)\}\&(i\%2=0)] \\
D*\{i-(C-I-1)\}+\{j-(D-J-1)\} & \text{when } [\{j\geq(D-J)\}\&(j\%2=0)][\{i\geq(C-I-1)\}(i\%2=1)] \\
D*(i+I+1)+\{j-(D-J+1)\} & \text{when } [\{j\geq(D-J)\}\&(j\%2=1)][\{i<(C-I-1)\}(i\%2=1)] \\
D*\{i-(C-I-1)\}+\{j-(D-J+1)\} & \text{when } [\{j\geq(D-J)\}\&(j\%2=1)]\&[\{i\geq(C-I-1)\}\&(i\%2=1)]
\end{cases} \tag{2}$$

$k_{n(64-QAM)}$

$$= \begin{cases}
D*(i+I)+(j+J) & \text{when } \{j<(D-J)\}\&[\{i<(C-I)\}\&(i\%3=0)] \\
D*(i+I)+(j+J+2) & \text{when } [\{j<(D-J)\}\&(j\%3=0)]\&[\{i<(C-I)\}\&(i\%3=1)] \\
D*(i+I)+(j+J+1) & \text{when } [\{j<(D-J)\}\&(j\%3\neq2)]\&[\{i<(C-I)\}\&(i\%3=2)] \\
D*\{i-(C-I)\}+(j+J) & \text{when } \{j<(D-J)\}\&[\{i\geq(C-I)\}\&(i\%3=0)] \\
D*\{i-(C-I)\}+(j+J+2) & \text{when } [\{j<(D-J)\}\&(j\%3=0)]\&[\{i\geq(C-I)\}\&(i\%3=1)] \\
D*\{i-(C-I)\}+(j+J+1) & \text{when } [\{j<(D-J)\}\&(j\%3\neq2)]\&[\{i\geq(C-I)\}\&(i\%3=2)] \\
D*(i+I)+(j+J-1) & \text{when } [\{j<(D-J)\}\&(j\%3\neq0)]\&[\{i<(C-I)\}\&(i\%3=1)] \\
D*\{i-(C-I)\}+(j+J-1) & \text{when } [\{j<(D-J)\}\&(j\%3\neq0)]\&[\{i\geq(C-I)\}\&(i\%3=1)] \\
D*(i+I)+(j+J-2) & \text{when } [\{j<(D-J)\}\&(j\%3=2)]\&[\{i<(C-I)\}\&(i\%3=2)] \\
D*\{i-(C-I)\}+(j+J-2) & \text{when } [\{j<(D-J)\}\&(j\%3=2)]\&[\{i\geq(C-I)\}\&(i\%3=2)] \\
D*(i+I+1)+\{j-(D-J)\} & \text{when } \{j\geq(D-J)\}\&[\{i<(C-I-1)\}\&(i\%3=0)] \\
D*(i+I+1)+\{j-(D-J-2)\} & \text{when } [\{j\geq(D-J)\}\&(j\%3=0)]\&[\{i<(C-I-1)\}\&(i\%3=1)] \\
D*(i+I+1)+\{j-(D-J-1)\} & \text{when } [\{j\geq(D-J)\}\&(j\%3\neq2)]\&[\{i<(C-I-1)\}\&(i\%3=2)] \\
D*\{i-(C-I-1)\}+\{j-(D-J)\} & \text{when } \{j\geq(D-J)\}\&[\{i\geq(C-I-1)\}\&(i\%3=0)] \\
D*\{i-(C-I-1)\}+\{j-(D-J-2)\} & \text{when } [\{j\geq(D-J)\}\&(j\%3=0)]\&[\{i\geq(C-I-1)\}\&(i\%3=1)] \\
D*\{i-(C-I-1)\}+\{j-(D-J-1)\} & \text{when } [\{j\geq(D-J)\}\&(j\%3\neq2)]\&[\{i\geq(C-I-1)\}\&(i\%3=2)] \\
D*(i+I+1)+\{j-(D-J+1)\} & \text{when } [\{j\geq(D-J)\}\&(j\%3\neq0)]\&[\{i<(C-I-1)\}\&(i\%3=1)] \\
D*\{i-(C-I-1)\}+\{j-(D-J+1)\} & \text{when } [\{j\geq(D-J)\}\&(j\%3\neq0)]\&[\{i\geq(C-I-1)\}\&(i\%3=1)] \\
D*(i+I+1)+\{j-(D-J+2)\} & \text{when } [\{j\geq(D-J)\}\&(j\%3=2)]\&[\{i<(C-I-1)\}\&(i\%3=2)] \\
D*\{i-(C-I-1)\}+\{j-(D-J+2)\} & \text{when } [\{j\geq(D-J)\}\&(j\%3=2)]\&[\{i\geq(C-I-1)\}\&(i\%3=2)]
\end{cases} \tag{3}$$

The general validity of the proposed mathematical formulation can be established with the help of *Asghar & Liu (2009)*. As far as spatial permutation is concerned, the steps involved in IEEE 802.16e (*Asghar & Liu, 2009*) and IEEE 802.11n (*IEEE 802.11n-2009, 2009*) are identical. Additionally, the latter undergoes frequency rotation using the frequency

**Table 3 Values of J$_{rot}$ for all modulation schemes, spatial streams and BWs.**

| Modulation scheme (N$_{cbpsc}$) | BW = 20 MHz | | | | BW = 40 MHz | | | |
|---|---|---|---|---|---|---|---|---|
| | I$_{ss}$ = 1 | I$_{ss}$ = 2 | I$_{ss}$ = 3 | I$_{ss}$ = 4 | I$_{ss}$ = 1 | I$_{ss}$ = 2 | I$_{ss}$ = 3 | I$_{ss}$ = 4 |
| BPSK (N$_{cbpsc}$ = 1) | 0 | 26 | 13 | 39 | 0 | 58 | 29 | 87 |
| QPSK (N$_{cbpsc}$ = 2) | 0 | 52 | 26 | 78 | 0 | 116 | 58 | 174 |
| 16-QAM (N$_{cbpsc}$ = 4) | 0 | 104 | 52 | 156 | 0 | 232 | 116 | 348 |
| 64-QAM (N$_{cbpsc}$ = 6) | 0 | 156 | 78 | 234 | 0 | 348 | 174 | 522 |

interleaving step, as described by B$_3$ in Fig. 1 for spatial streams other than the first. Further, analysis of the third step results that the entire term beyond $j_k$ (i.e., $J_{rot}$) remains constant for a particular spatial stream and expressed by Eq. (4) (*Asghar & Liu, 2009*).

$$r_k = [j_k - J_{rot}]\%N \qquad (4)$$

where $J_{rot} = \left[ \{(i_{ss} - 1) * 2\}\%3 + 3\left\lfloor \frac{i_{ss}-1}{3} \right\rfloor \right] * N_{rot} * N_{BPSCS}$

As the first stream for all modulation schemes undergoes no frequency rotation, hence

$$r_k = [j_k - 0]\%N = [j_k]\%N = j_k$$

For subsequent streams, the value of $J_{rot}$ differs for each spatial stream, modulation schemes, and BWs. All such possible values of $J_{rot}$ are listed in Table 3. The expression of $j_k$ so derived for all modulation schemes in *Asghar & Liu (2009)* if substituted in Eq. (1) gives three new equations. The final expressions obtained and the proposed mathematical formulations developed in this work generate the same results and are identical to results obtained through direct implementation of B$_1$ to B$_3$ steps.

The work reported in this paper includes interleaver design for all four modulation schemes (i.e., BPSK, QPSK, 16-QAM, and 64-QAM) as defined in the IEEE 802.11n standard. However, the proposed algorithm may be generalized, as follows, to include any other modulation scheme beyond the above standard.

1. Define the number of coded bits per sub-carrier ($N_{bpscs}$) for the modulation scheme beyond the above standard and compute $s = max\ (1, N_{bpscs})$.
2. Define interleaver depth ($N$), number of columns ($d$), and compute the intermediate addresses after spatial interleaving ($j_k$) by implementing B$_1$–B$_2$ steps.
3. Compute the final memory addresses ($r_k$) by implementing step B$_3$ with appropriate values of frequency rotation parameter ($N_{rot}$) corresponding to the permissible bandwidths (BWs) for all four values of spatial streams ($i_{ss}$).
4. Arrange the addresses obtained in step 3 in ($N/N_{rot}$) x $d$ tabular form with $j$ and $i$ as row and column numbers, respectively.
5. Identify the correlation between the subsequent addresses and re-arrange each address in ($N/N_{rot}$) * ($i \pm offset_1$) + ($j \pm offset_2$) format. The $offset_x = 0$ with $i_{ss} = 1$ for all values of $N$. All other values of $offset_x$ to be computed using the correlation between the subsequent addresses.

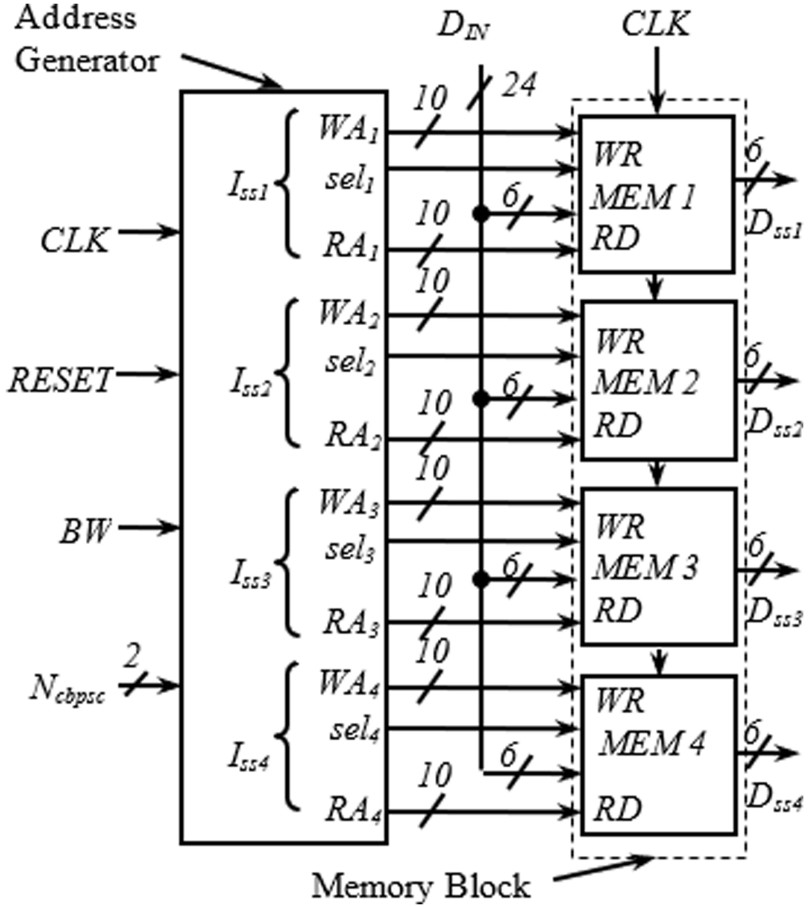

**Figure 2 Top level view of complete interleaver.**

6. Group the above mathematical expressions (obtained from step 5) according to the specific modulation scheme. These expressions exclude floor function, hence, suitable for implementation on the hardware platform.

7. Functional verification of the algorithm may be carried out by comparing the addresses of steps 3 and 6 using suitable software.

## TRANSFORMATION INTO HARDWARE

This section describes the transformation of the proposed address generator algorithm into digital hardware. The top-level view of the complete interleaver consisting of the proposed address generator and memory block is shown in Fig. 2.

### Memory block

The detailed arrangement of the memory block for one spatial stream having a similar structure as in *Upadhyaya & Sanyal (2011)* is shown in Fig. 3. The structure is generic and applies to all spatial streams. It receives three inputs from the address generator block; write address ($WA_x$), read address ($RA_x$), and $sel_x$. The requirement of two memory blocks for block interleaving is accomplished here with a dual-port memory (with Ports A and B)

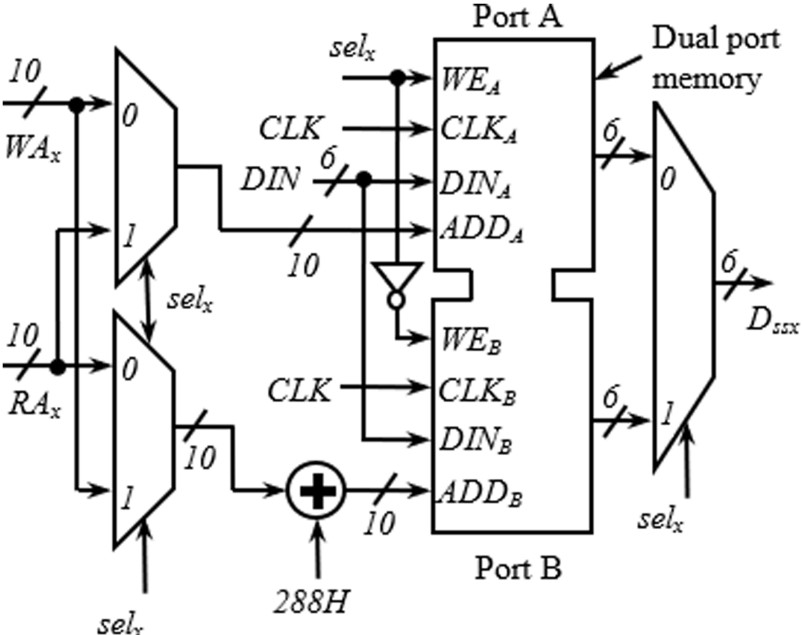

**Figure 3 Internal structure of memory block.**

where the read and write operations can be performed simultaneously. As seen in Fig. 3, the first 288H locations are used as Port A, and the next 288H locations as Port B. An adder is used to insert the bias of 288H while generating the addresses for Port B. When one port is being written, another one is read, and vice versa. Swapping between read/write operations at the end of a cycle is performed using the signal $sel_x$, generated using a toggle flip flop.

## Address generator

The address generator is the heart of the interleaver. The encoding schemes used in this work for the two inputs, $BW$ and $N_{cbpsc}$ of the address generator, are described in Table 4. The $i_{ss1}$–$i_{ss4}$ represent the four different spatial streams of the address generator, each consisting of write ($WA_x$), read ($RA_x$) address, and select signal ($sel_x$) output. As shown in Fig. 4, in the write address generator, a multiplexer is used to route the desired $WA_x$ from four possible sources based on the value of $N_{cbpsc}$ for a particular spatial stream, $I_{ssx}$.

Figures 5A and 5B show the hardware used for the generation of row-count ($JCOUNT$) and column-count ($ICOUNT$), respectively, which consist of up-counters and comparators. As per (*IEEE 802.11n-2009, 2009*), the column number is defined as $C = 13$ and 18, for BW = 20 and 40 MHz, respectively. Circuit arrangement for the generation of row number, $D$ using $BW$ and $N_{cbpsc}$ is shown in Fig. 6. Similarly, Figs. 7A and 7B describe hardware used for the generation of $ICOUNT < (C - I_x)$, $ICOUNT \geq (C - I_x)$, $JCOUNT < (D - J_x)$, and $JCOUNT \geq (D - J_x)$ signals. Here $I_x$ and $J_y$ are the column and row offset values, respectively, which are used while computing the addresses, and are defined in Table 5.

**Table 4  Encoding of BW.**

**(A) Encoding of BW**

| Bandwidth ($BW$) | Encoded bit |
|---|---|
| 20 MHz | 0 |
| 40 MHz | 1 |

**(B) Encoding of $N_{cbpsc}$**

| Modulation scheme ($N_{cbpsc}$) | Encoded bits |
|---|---|
| BPSK ($N_{cbpsc} = 1$) | 00 |
| QPSK ($N_{cbpsc} = 2$) | 01 |
| 16-QAM ($N_{cbpsc} = 4$) | 10 |
| 64-QAM ($N_{cbpsc} = 6$) | 11 |

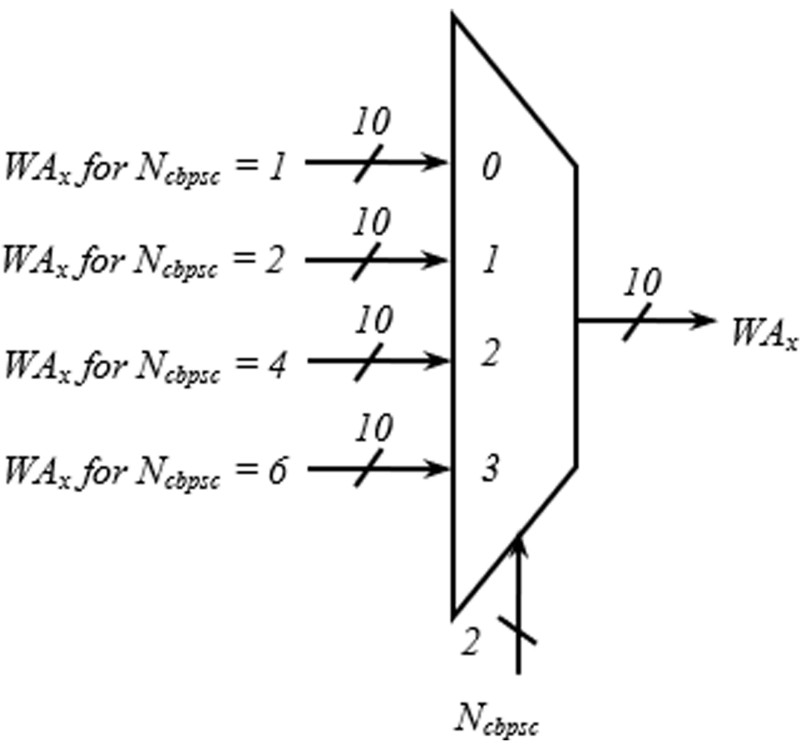

**Figure 4  Multiplexing scheme showing write address generation for one spatial stream.**

The hardware required for the generation of $RA_x$ is shown in Fig. 8. Like the write address generator, the structure developed for the generation of $RA_x$ is also generic and is applicable to all the spatial streams. The first and second level multiplexers select one of the values of interleaver depth from the inputs with $BW$ and $mod\_typ$ signal. The $rd\_count$ is a 10-bit up counter and generates $RA_x$. While progressing through the count values,

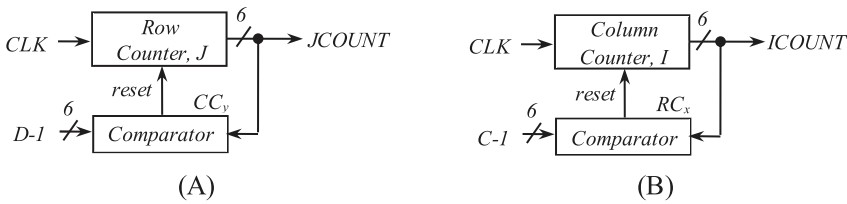

**Figure 5** Scheme showing generation of (A) row count and (B) column count.

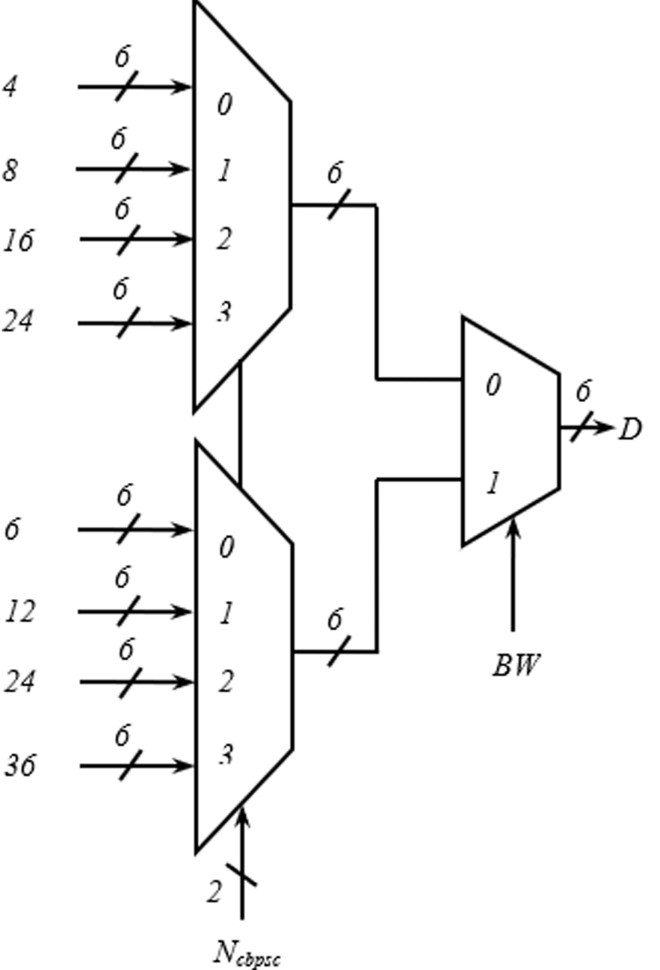

**Figure 6 Scheme for generation of number of rows (D).**

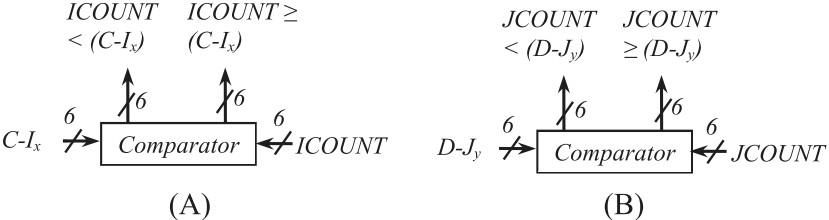

**Figure 7 Arrangement showing generation of (A) ICOUNT < (C − Iₓ) and ICOUNT ≥ (C − Iₓ)** **(B) JCOUNT < (D − Jᵧ) and JCOUNT ≥ (D − Jᵧ).**

**Table 5 Definition of $I_x$ and $J_y$ for all streams and BW.**

| Stream | BW = 20 MHz, C = 13 | BW = 40 MHz, C = 18 |
|---|---|---|
| $I_{ss1}$ | $I_1 = 0, J_1 = 0$ | $I_1 = 0, J_1 = 0$ |
| $I_{ss2}$ | $I_2 = 6, J_2 = NBPSC * 2$ | $I_2 = 8, J_2 = NBPSC * 2$ |
| $I_{ss3}$ | $I_3 = 9, J_3 = NBPSC * 3$ | $I_3 = 13, J_3 = NBPSC$ |
| $I_{ss4}$ | $I_4 = 3, J_4 = NBPSC$ | $I_4 = 3, J_4 = NBPSC * 3$ |

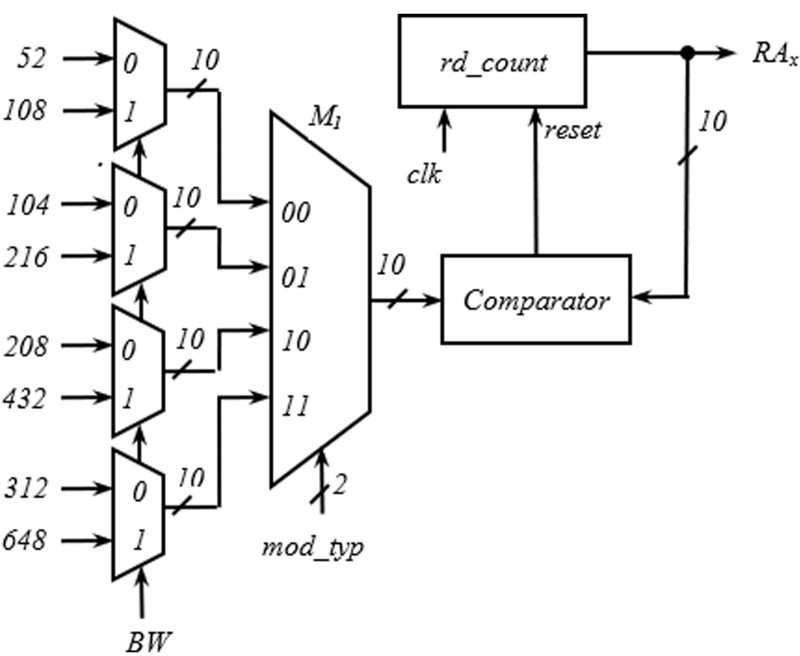

**Figure 8 Circuit for generation of read address ($RA_x$).**

when the *rd_count* value equals the output of $M_1$, a reset pulse is generated by the comparator, and rd_count goes to the initial state to start another cycle.

Figures 9 and 10 show the rest of the circuit details required to generate $WA_x$ with BPSK/QPSK, 16-QAM, and 64-QAM modulation schemes. In these figures, the adders ($A_1$, $A_2$, and $A_3$) receive two inputs; one from the row-count part (purple colored) and the other from the column-count part (blue colored) of the circuit. In Fig. 9, the $JCOUNT + J_y$ signal is generated by an adder ($A_4$), whereas the two subtractors ($S_1$ and $S_2$) generate the signal $JCOUNT - (D - J_y)$. Based on the value of $JCOUNT < (D - J_y)$ signal, the multiplexer ($M_2$) routes one of these signals to the input of the $A_1$. Similar hardware structures can be found to generate signals like $ICOUNT + I_x$, $ICOUNT + I_x + 1$, $ICOUNT - (C - I_x)$, etc., in the column-count part. The column-count part's output gets multiplied with $D$ in the multiplier ($ML_1$) to generate the second input of $A_1$. In Fig. 10, the circuit details for generating signals like $ICOUNT + I_x$, $ICOUNT - (C - I_x)$, $JCOUNT + J_y$, $JCOUNT - (D - J_y)$, etc., are not shown to avoid repetition and clumsiness. The condition for the generation of select inputs (II4, JJ4, II6, and JJ6) for the multiplexers of Fig. 10, are described and encoded in Tables 6(A) and (B).

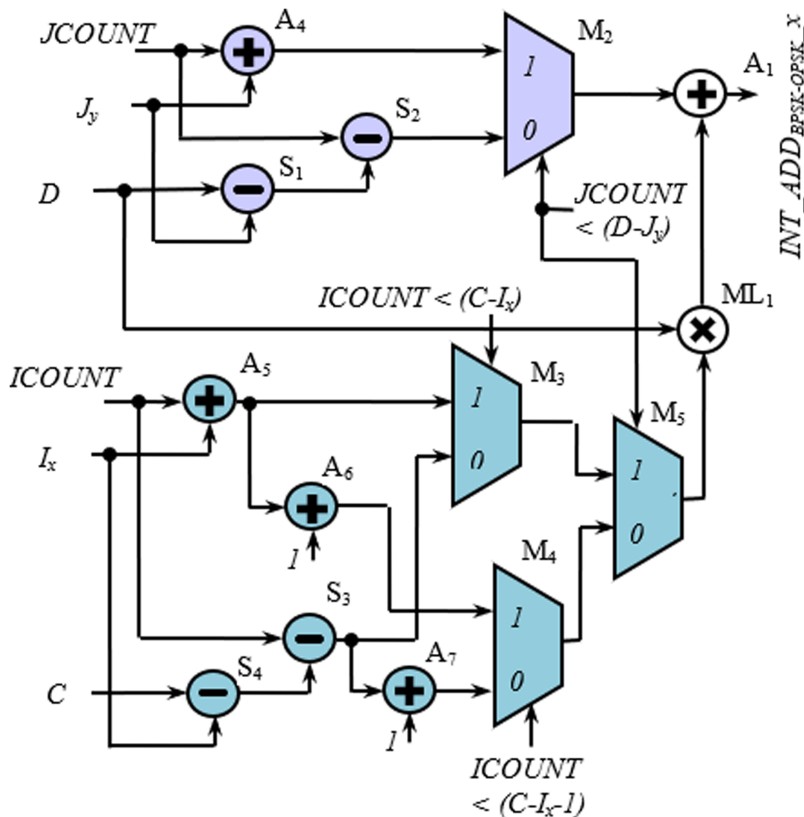

**Figure 9  Circuit diagram for generation of interleaver write addresses with $N_{cbpsc}$ = 1 or 2.**

**Table 6  Encryption of signals.**

**(A) Encryption of signals II4 and JJ4**

| Condition | II4 | Condition | JJ4 |
|---|---|---|---|
| $ICOUNT < (C − I_x)$ and $iXMOD = 0$ | 00 | $JCOUNT < (D − J_y)$ and $jXMOD = 0$ | 00 |
| $ICOUNT < (C − I_x)$ and $iXMOD = 1$ | 01 | $JCOUNT < (D − J_y)$ and $jXMOD = 1$ | 01 |
| $ICOUNT ≥ (C − I_x)$ and $iXMOD = 0$ | 10 | $JCOUNT ≥ (D − J_y)$ and $jXMOD = 0$ | 10 |
| $ICOUNT ≥ (C − I_x)$ and $iXMOD = 1$ | 11 | $JCOUNT ≥ (D − J_y)$ and $jXMOD = 1$ | 11 |

**(B) Encryption of signals II6 and JJ6**

| Condition | II6 | Condition | JJ6 |
|---|---|---|---|
| $ICOUNT < (C − I_x)$ and $iXMOD = 0$ | 000 | $JCOUNT < (D − J_y)$ and $jXMOD = 0$ | 000 |
| $ICOUNT < (C − I_x)$ and $iXMOD = 1$ | 001 | $JCOUNT < (D − J_y)$ and $jXMOD = 1$ | 001 |
| $ICOUNT < (C − I_x)$ and $iXMOD = 2$ | 010 | $JCOUNT < (D − J_y)$ and $jXMOD = 2$ | 010 |
| $ICOUNT ≥ (C − I_x)$ and $iXMOD = 0$ | 011 | $JCOUNT ≥ (D − J_y)$ and $jXMOD = 0$ | 011 |
| $ICOUNT ≥ (C − I_x)$ and $iXMOD = 1$ | 100 | $JCOUNT ≥ (D − J_y)$ and $jXMOD = 1$ | 100 |
| $ICOUNT ≥ (C − I_x)$ and $iXMOD = 2$ | 101 | $JCOUNT ≥ (D − J_y)$ and $jXMOD = 2$ | 101 |

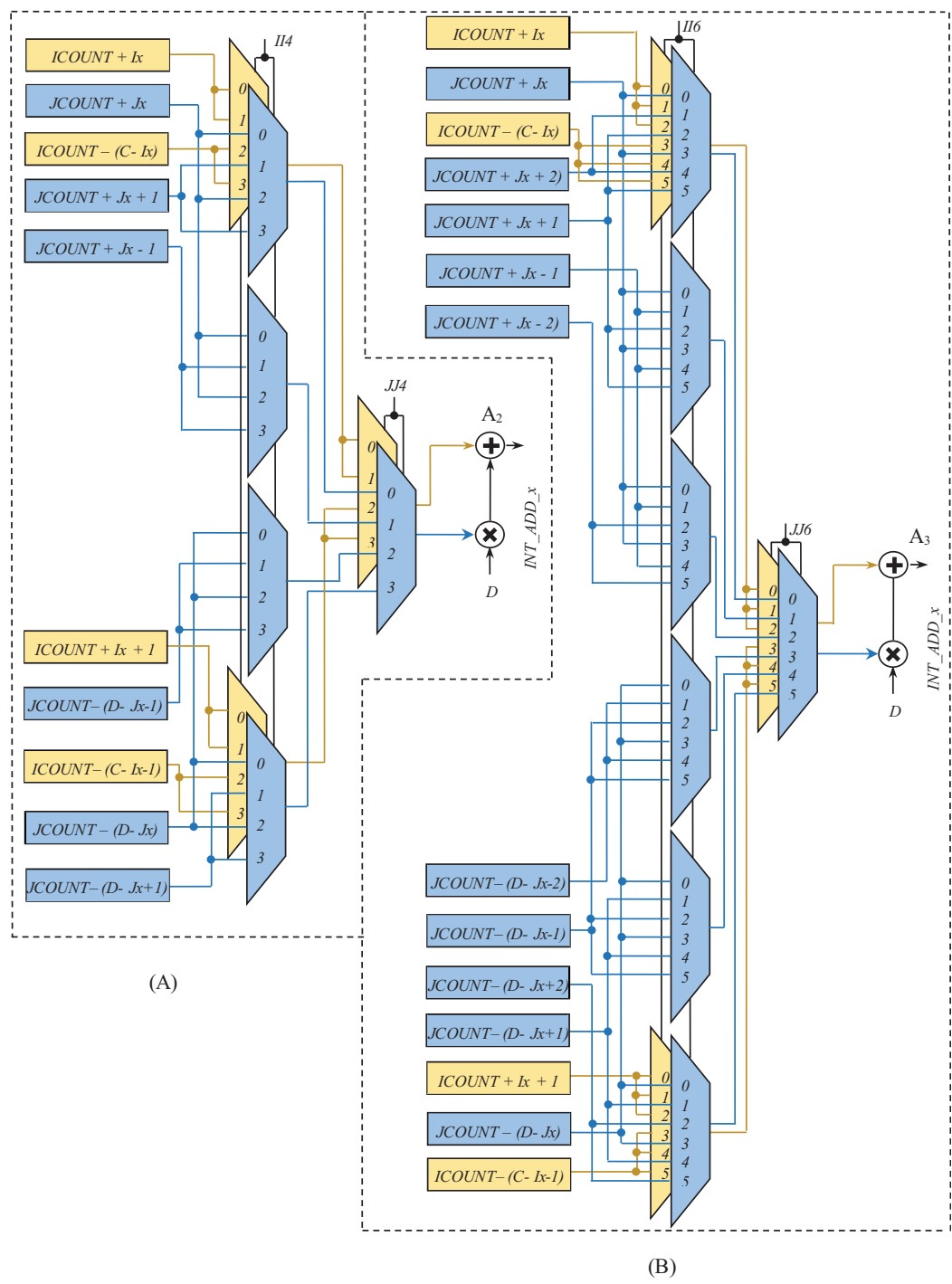

**Figure 10 Circuit diagram for generation of interleaver write addresses with (A) N_cbpsc = 4 and (B) N_cbpsc = 6.**

## SIMULATION RESULTS

The digital hardware of the MIMO WLAN interleaver is translated into a VHDL program using Xilinx ISE 12.1. The proposed design of the interleaver is simulated, and the functionality verification is done using ModelSim XE-III. The address generation circuitry

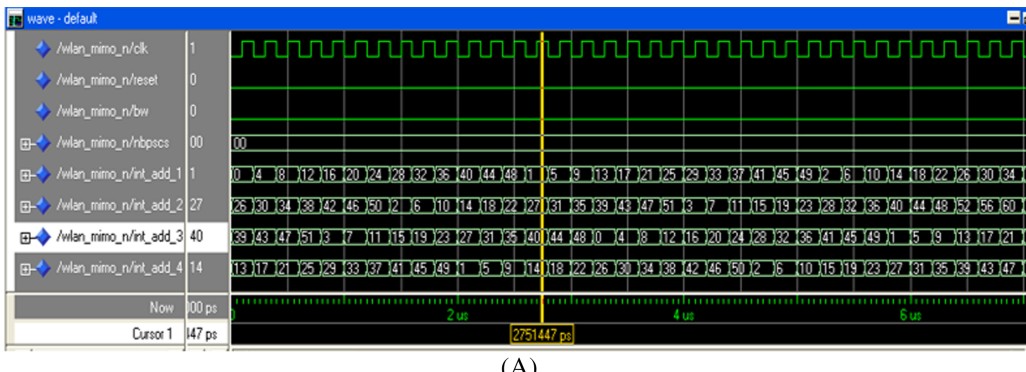

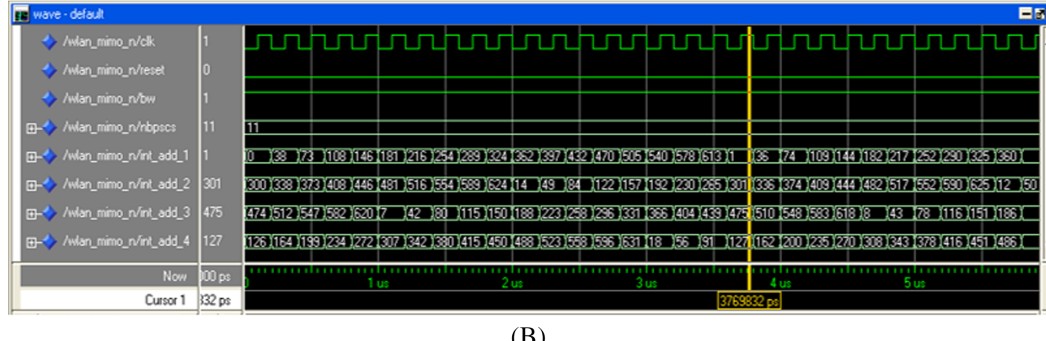

**Figure 11 Write addresses (WA$_x$) for (A) BW = 0 (20 MHz), N$_{bpscs}$ = 00 (BPSK), (B) BW = 1 (40 MHz), N$_{bpscs}$ = 11 (64-QAM).**

of the interleaver is tested for all BWs, spatial streams, and modulation schemes, out of which two results (for $BW = 0$, $N_{bpscs} = 00$ and $BW = 1$, $N_{bpscs} = 11$) are presented in Figs. 11A, 11B. The last four signals (int_add_1 to int_add_4) of Figs. 11A and 11B show the sequence of write addresses generated in synchronization with a clock signal (clk) for all the four spatial streams of the interleaver ($I_{ss1} - I_{ss4}$). The write address sequence generated by the proposed interleaver for spatial stream 1 (i.e., int_add_1) is 0, 4, 8, 12, … Similarly, the address sequence for spatial stream 2 (i.e., int_add_2) is 26, 30, 34, and so on. The last address sequence (i.e., int_add_4) of Fig. 11A tallies with the address sequences shown in Table 1(A). Automatic address verification has also been carried out between the addresses generated by our proposed algorithm and the addresses obtained through steps B$_1$–B$_3$ of "Interleaving in IEEE 802.11n" involving floor function by running a separate MATLAB program. This verification further endorses the correctness of the proposed algorithm.

# FPGA IMPLEMENTATION RESULTS

The proposed design of the interleaver is transformed into a VHDL model using Xilinx ISE 12.1 and is implemented on Xilinx Spartan-6 FPGA. Despite our exhaustive literature survey, any similar implementation on the FPGA platform has not been noticed. As a result, the conventional LUT-based approach has been implemented on the same FPGA platform utilizing Block RAM (BRAM) to house the address LUTs. Four dual port BRAM memory blocks are used to implement the interleaver memory in both designs.

**Table 7 Device Utilization Summary.**

| FPGA resources | This work | | LUT based technique | |
| --- | --- | --- | --- | --- |
| | Utilization in number | Utilization in % | Utilization in number | Utilization in % |
| Number of slices registers | 30 out of 30,064 | 0.10 | 35 out of 30,064 | 0.12 |
| Number of slices LUTs | 864 out of 15,032 | 5.75 | 201 out of 15,032 | 1.34 |
| Number of BRAMs | 4 out of 52 | 7.69 | 36 out of 52 | 69.23 |
| Number of DSP48A1s | 4 out of 38 | 10.53 | 0 out of 38 | 0 |
| Number of BUFG/BUFGCTRLs | 2 out of 16 | 12.50 | 2 out of 16 | 12.50 |

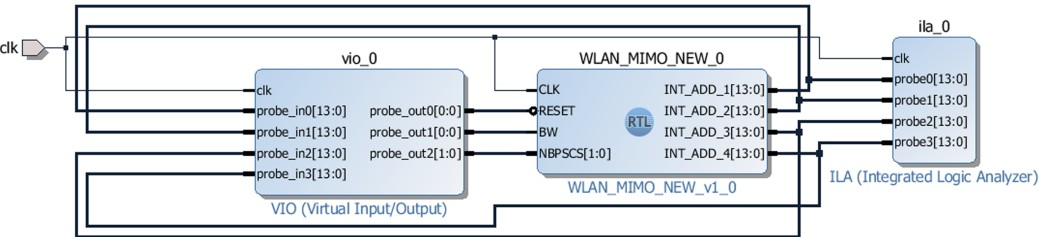

**Figure 12 Test arrangement of the address generator using VIO and ILA.**

Comparative analysis of the two implementations in terms of device utilization is made in Table 7. The betterment of the proposed technique can be quantified in terms of embedded memory utilization (88.9% memory block saving) and operating speed (37.8% speed improvement). The use of DSP blocks as multiplier improves the performance of the circuit by reducing delay. The circuit works at a maximum clock frequency (*f*) of 208.7 MHz with 28.62 mW of total power consumption, which includes static and dynamic power. The use of FPGA's embedded DSP blocks (DSP48A1s) as a multiplier and embedded dual-port memory (BRAM) helps to reduce the memory access time and, in turn, improves the throughput of the system. Resource efficiency and compact design are the key contributors to reducing the power consumption of the interleaver. In addition, Spartan 6 FPGA itself is known for better power efficiency, increased productivity, and higher performance implementation platform.

The hardware testing of the address generator for the MIMO WLAN interleaver has been performed using VIO and ILA. VIO and ILA are the customizable cores that facilitate both monitoring and driving internal FPGA signals in real-time. Figure 12 shows the block level design of the test environment using VIO and ILA wherein the proposed address generator block (WLAN_MIMO_NEW_0) is placed in the middle of the VIO (left side) and ILA (right side) blocks. The VIO injects user-defined RESET, BW, and NBPSCS signals. The outputs generated by the address generator (INT_ADD_1, 2, 3, and 4) are fed to the ILA and VIO for verification. An external clock (clk) signal drives all the modules synchronously.

The throughputs of the proposed interleaver for all four modulation schemes are computed using Eq. (5) and presented in Table 8.

**Table 8 Throughput comparison with IEEE 802.11n.**

| Maximum throughput requirement of IEEE 802.11n | This work | | |
| --- | --- | --- | --- |
| | Modulation scheme | Maximum throughput | Improvement over IEEE 802.11n |
| 600 Mbps | BPSK | 834.8 Mbps | 1.39 times |
| | QPSK | 1669.6 Mbps | 2.78 times |
| | 16-QAM | 3339.2 Mbps | 5.57 times |
| | 64-QAM | 5008.8 Mbps | 8.35 times |

**Table 9 Comparative study with similar works.**

| FPGA parameters | This work | (*Zhang et al., 2009*) | (*Asghar & Liu, 2009*) | (*Zhang et al., 2012*) | LUT based |
| --- | --- | --- | --- | --- | --- |
| Maximum clock frequency, $f$ | 208.7 MHz | 109.38 MHz | 70.31 MHz | 125 MHz | 151.45 MHz |
| Power consumption, $P$ | 28.62 mW | 111.24 mW | 48 mW | Not available | 28.62 mW |

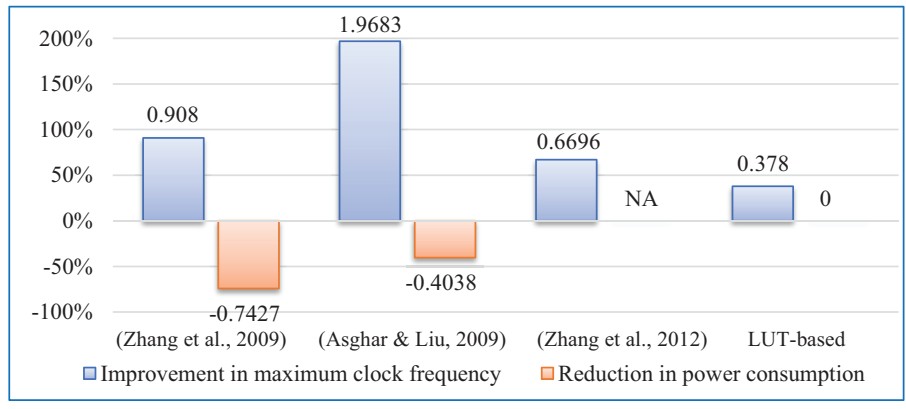

**Figure 13 Performance comparison.** 

$$Tp = f \times N_{bpscs} \times i_{ss} \tag{5}$$

The last column of Table 8 justifies our high throughput claim of the proposed interleaver. This provides the opportunity to implement the proposed design in relatively slower and lower-cost FPGAs as well, thereby providing a cost-effective solution.

Besides, a comparison with few works has been made based on the equivalence drawn between FPGA and ASIC implementations in *Kuon & Rose (2006)*. The comparative study of the proposed implementation regarding key FPGA parameters shows betterment over other similar recent works and is presented in Table 9 and Fig. 13. The proposed circuit shows betterment over (*Asghar & Liu, 2009*; *Zhang et al., 2009*; *Zhang et al., 2012*), and LUT-based technique in terms of maximum operating frequency. In terms of power consumption, our implementation is found to be the most efficient among the designs presented in *Asghar & Liu (2009)*, *Zhang et al. (2009)*, *Zhang et al. (2012)* of Fig. 13.

As direct implementation of floor function is not possible, improvement in terms of memory block used (BRAM) and clock frequency over the LUT-based technique may be considered as the performance improvement of our novel algorithm due to the elimination of floor function from the interleaver address generator circuitry.

Massive MIMO system, a key technology being deployed in the 5G system, employs an array of a large number of transmitting antennas at the base station to achieve high throughput has been investigated to compare our FPGA implementation results. *Tang, Chen & Zhang (2016)* have demonstrated CMOS implementation of the message-passing detector (MPD) designed for a 256-QAM massive MIMO system supporting 32 concurrent mobile users in each time-frequency resource with 2.76 Gbps throughput. As far as throughput is concerned, our proposed interleaver on the FPGA platform shows a competitive result with that of *Tang, Chen & Zhang (2016)*.

## CONCLUSIONS

This work demonstrates the design and implementation of novel interleaver hardware on the FPGA platform to be used in OFDM-based MIMO WLAN applications. A new algorithm has been proposed for the address generator of the interleaver eliminating the requirement of floor function, and is supported by the mathematical formulation with general validity. The algorithm is transformed into the digital circuit and is modeled using VHDL software. Simulation results and hardware testing verify the functionality of the proposed algorithm. Hardware implementation of the VHDL model using Xilinx ISE is done and is tested on Xilinx Spartan 6 FPGA. Efficient design and use of FPGA's embedded resources during implementation enables betterment over a few recent similar works and conventional design in terms of multiple FPGA parameters and the interleaver throughput.

### Funding
The authors received no funding for this work.

### Competing Interests
The authors declare that they have no competing interests.

### Author Contributions
- Bijoy Kumar Upadhyaya conceived and designed the experiments, performed the experiments, analyzed the data, performed the computation work, prepared figures and/or tables, authored or reviewed drafts of the paper, and approved the final draft.
- Pijush Kanti Dutta Pramanik analyzed the data, prepared figures and/or tables, authored or reviewed drafts of the paper, and approved the final draft.
- Salil Kumar Sanyal analyzed the data, authored or reviewed drafts of the paper, and approved the final draft.

## Data Availability
The coding of the simulation experiment is available as a Supplemental File.

## Supplemental Information
Supplemental information for this article can be found online at http://dx.doi.org/10.7717/peerj-cs.581#supplemental-information.

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
