# Peer review of "High throughput resource efficient reconfigurable interleaver for MIMO WLAN application"

_PeerJ Computer Science, doi:10.7717/peerj-cs.581_

## Round 0.1 · original submission · Major Revisions

The paper requires a major revision to address reviewer comments before publication.

Reviewer 1 ·

Basic reporting

In this work, the authors propose a novel algorithm to efficiently model the address generation circuitry of MIMO WLAN interleaver. The interleaver used in MIMO WLAN transceiver has three steps of permutation involving floor function whose hardware implementation is considered to be most challenging due to the absence of corresponding digital hardware. Experimental results are promising, comparisons with few recent similar works including the conventional Look-Up Table (LUT) based technique demonstrated the superiority of the proposed design in terms of maximum improvement in operating frequency by 196.83%, maximum reduction in power consumption by 74.27% and reduction of memory occupancy by 88.9%.

Experimental design

Without comments

Validity of the findings

1 - More comparisons with previous works could be a nice complement for the current manuscript. Then, the authors could demonstrate how the proposed approach outperform the current state of the art.
2 - A post-synthesis simulation could be a nice complement for the current manuscript.

Additional comments

In general, the manuscript is interesting, well organized and deserves to be published. There are a little grammatical/style error. In my opinion, a grammar/style revision has to be carried out before the manuscript can be considered for publication.

·

Basic reporting

In this work, according to the authors, they propose an algorithm with the mathematical background for the address generator eliminating the need for floor function. The algorithm is converted into digital hardware for implementation on the reconfigurable FPGA platform. Hardware structure for the complete interleaver including the reading address generator and the memory module is designed and modeled in VHDL using Xilinx Integrated Software Environment (ISE) utilizing embedded memory and DSP blocks of the target FPGA. in the end, the functionality of the proposed algorithm is verified through exhaustive software simulation. The authors has done work for high-speed communication which is an appreciate able work domain.

I have a concern that can be helpful to improve the quality of the manuscript.

Experimental design

1- Experimental approach of the study is not clear, the author should summarize it in a clear pattern so it can be easy for the reader to understand it.

2- In the abstract, the author mentions that they verified the functionality of the proposed algorithm through exhaustive software simulation, please elaborate it why? is there any other method to verify? if yes then why not others?

Validity of the findings

no comment

Additional comments

1- The major issue is the presentation of content it should be better and understandable.
2- English grammar and typos should be checked thoroughly.
3- Some keypoint should be add in the introduction section for the ease of the reader to understand the article's contribution.

Reviewer 3 ·

Basic reporting

.

Experimental design

.

Validity of the findings

.

Additional comments

Highlights the main scientific contributions in section 1.

Section 5 and 6 are the main section of the paper, however they are extremely poor. Authors must
insert more information and results.

Table 8:
If the maximum throughput requirement of IEEE 802.11n is 600 Mbps which is advantage in work
with more? The authors must clarify this information.

Table 9:
The power consumption increases non-linearly with frequency cube (P ≈ F3). How, does the proposal have small consumption then literature? The authors must clarify this information.
Is this information dynamic or static power?

---

## Round 0.2 · Minor Revisions

Please look at the following comment "typos are still a problem in the manuscript".

·

Basic reporting

The author proposes an algorithm to efficiently model the address generation circuitry of the MIMO WLAN interleaved. The interleaved used in the MIMO WLAN transceiver has three permutation steps involving floor function whose hardware implementation is most challenging due to the absence of corresponding digital hardware. They proposed an algorithm with a mathematical background for the address generator, eliminating the need for floor function. The algorithm is converted into digital hardware for implementation on the reconfigurable FPGA platform. Hardware structure for the complete interleaved, including the read address generator and memory module, is designed and modeled in VHDL using Xilinx Integrated Software Environment (ISE) utilizing embedded memory and DSP blocks Spartan 6 FPGA.

Author work in a good manner in revision on raised point and tackle the things very well.

Experimental design

The experimental explanation and flow are cleared in revision.

Validity of the findings

The author presents results well which improves the quality of the manuscript.

Additional comments

The manuscript is much strong after revision but typos are still a problem in the manuscript.

---

## Round 0.3 · accepted · Accept

Thank you for making the amendments. I am pleased to recommend the paper for acceptance.